# Fabrication of SnO_2_ Composite Nanofiber-Based Gas Sensor Using the Electrospinning Method for Tetrahydrocannabinol (THC) Detection

**DOI:** 10.3390/mi11020190

**Published:** 2020-02-12

**Authors:** Pouria Mehrabi, Justin Hui, Sajjad Janfaza, Allen O’Brien, Nishat Tasnim, Homayoun Najjaran, Mina Hoorfar

**Affiliations:** School of Engineering, University of British Columbia, Kelowna, BC V1V 1V7, Canada; pouria.mehrabi91@gmail.com (P.M.); huijustin7@gmail.com (J.H.); sajad.janfaza@gmail.com (S.J.); aobrien1@ualberta.ca (A.O.); nishat11@mail.ubc.ca (N.T.); homayoun.najjaran@ubc.ca (H.N.)

**Keywords:** gas sensor, electrospinning, SnO_2_ nanofibers, gold functionalization

## Abstract

This paper presents the development of a metal oxide semiconductor (MOS) sensor for the detection of volatile organic compounds (VOCs) which are of great importance in many applications involving either control of hazardous chemicals or noninvasive diagnosis. In this study, the sensor is fabricated based on tin dioxide (SnO_2_) and poly(ethylene oxide) (PEO) using electrospinning. The sensitivity of the proposed sensor is further improved by calcination and gold doping. The gold doping of composite nanofibers is achieved using sputtering, and the calcination is performed using a high-temperature oven. The performance of the sensor with different doping thicknesses and different calcination temperatures is investigated to identify the optimum fabrication parameters resulting in high sensitivity. The optimum calcination temperature and duration are found to be 350 °C and 4 h, respectively and the optimum thickness of the gold dopant is found to be 10 nm. The sensor with the optimum fabrication process is then embedded in a microchannel coated with several metallic and polymeric layers. The performance of the sensor is compared with that of a commercial sensor. The comparison is performed for methanol and a mixture of methanol and tetrahydrocannabinol (THC) which is the primary psychoactive constituent of cannabis. It is shown that the proposed sensor outperforms the commercial sensor when it is embedded inside the channel.

## 1. Introduction

The global gas sensor market size is evaluated at $1 billion and is predicted to reach $5 billion by 2020 [1]. There is a wide range of industries applying gas sensors for monitoring and diagnosis, for example, in healthcare (for early detection of noncommunicable diseases [2,3,4,5] to reduce annual premature mortality rates by 25% according to World Health Organization report [6]); agriculture (for monitoring food quality, detecting crop diseases and insect infestations [7]); defence (for detection of explosives and chemical warfare agents [8]); and environment (for detection of VOC pollutants accounting for 37% (693 kilotons/year) of total emission according to Environment and Climate Change Canada report [9]). Highly sensitive gas detectors are essential, especially for applications involving detection of toxic gases and disease diagnosis. For instance, arsine gas, often used in the manufacturing of semiconductors, metal smelting, and extraction is deadly above an exposure threshold of 0.05 ppm, according to the Association Advancing Occupational and Environmental Health [10]. Highly sensitive gas sensors are required to detect toxic vapors (such as arsine gas) and prevent unnecessary injuries or death. For diagnosis applications, several studies have shown differences in concentration of volatile organic compounds (VOCs) in exhaled breath between healthy people and people with a particular disease [11]; these changes are often on the scale of parts per billion (ppb). For these reasons highly sensitive gas sensors would be an invaluable tool for lethal gas detection and early, noninvasive disease diagnosis.

Recently, breath analysis has gained interest as a noninvasive method for the detection of chemicals (or mixtures of chemicals) that are either produced as part of biological processes or are absorbed from the environment. For instance, smokers can be recognized from nonsmokers by analyzing their VOCs [11]. The detection of THC in the breath of smokers has attracted significant attention in recent years, and hence a few portable devices for detection of THC in the breath of marijuana smokers have recently been proposed [12]. In addition to breath analysis, recent legalization of marijuana in certain countries has created a vast market regarding cannabinoids extraction from marijuana (e.g., for medical and food industry). For quality assurance, the concentration of these components needs to be measured accurately.

Metal oxide based gas sensors are becoming increasingly popular due to their high sensitivity. Their highly sensitive gas sensing layer has been developed through miniaturization, providing nanoscale topology to increase surface reaction sites. A variety of sensing layer structures have been developed, such as, but not limited to, nanoparticles, nanofibers, nanorods, and nanosheets [13,14,15,16,17,18,19]. While each of these new developments has the common advantage of increasing the selectivity and sensitivity of metal oxide gas sensors, the major distinction comes from the fabrication process. Sensing layers containing nanorods and nanosheets require extensive processes such as high temperatures or special equipment for fabrication [16,17], while nanoparticles and nanofibers require relatively simple processes such as electrospinning [14,15]. In addition to these developments which provide significant advances in low concentration gas detection, further manufacturing processes have been developed to enhance both selectivity and sensitivity of a sensor. For instance, a popular method applied to increase the sensitivity and selectivity of semiconductor metal oxide sensors is through doping [20,21,22]. The performance enhancement based on doping was highlighted in a study completed by Xiang et al. [23] who doped ZnO nanorods with Ag. This study demonstrated the effectiveness of doping by exposing the doped nanorods and bare nanorods to 10 ppm of ethanol at different working temperatures. The study results showed that the doped nanorod sensor had higher response as compared with the undoped nanorod at each working temperature. This phenomenon was explained by Yamazoe et al. [24]; the catalytic properties of noble metals increase the rate of oxidation, thereby enhancing sensitivity. The challenging part is to find an effective doping method that can be used to introduce the dopants to the sensing material. Several methods have been used such as mixing [25], dip coating [26], and RF-magnetron/reactive DC sputtering [27].

In this paper, the DC sputtering of Au is used to enhance the performance of electrospun SnO_2_ calcinated nanofibers. The effect of calcination (for various temperatures) and Au doping (for various sputtering thicknesses) are studied to enhance the sensitivity of the sensor. Later on, the in-house fabricated sensor is embedded inside a microfluidic channel for better selectivity results. The in-house fabricated sensor shows a more sensitive/selective response to the THC solution as compared with the commercial MOS sensor.

## 2. Materials and Methods

The precursor solution includes a mixture of SnCl4, poly(ethylene oxide), deionized (DI) water, isopropanol, 1-propanol, and chloroform (all purchased from Sigma-Aldrich). First, 1 g of SnCl4 was dissolved in a solution containing 2 g of DI water, 0.6 g of propanol, and 0.4 g of isopropanol. The solution was stirred magnetically for 1 h at 50 °C to ensure that SnCl4 was well dissolved. Then, a solution of 125 mg of poly(ethylene oxide) and 6.25 mL of chloroform was added and stirred magnetically for another hour.

All the silicone substrates were first cleaned with piranha solution, rinsed with DI water and, then, dried using air flow at room temperature. The interdigitated fingers with 20 µm gaps were fabricated on the silicon substrate using the following microfabrication technique: 1 mm thick silicon wafer substrate was coated by sputtering 35 nm chrome followed by 65 nm gold (both at a rate of 1 Angstrom/sec). The S1813 photoresist solution was, then, spin coated at 2000 rpm for 30 sec, resulting in a 1 µm film thickness. To pattern the interdigitated electrodes, the masks were designed in which the 100 µm thick fingers were placed with a 20 µm gap size (Figure 1a). The photoresists film was cured on a hot plate at 115 °C for 60 s. The same procedure was performed on the back side of the silicon wafer to fabricate the microheater, as illustrated in Figure 1b. To transfer the patterns, UV illuminated the masks which were placed on the photoresist film. Then, the exposed regions (corresponding to transparent areas in the mask) were washed out by immersing the substrate into photoresist remover. Subsequently, the gold and chrome layers at those regions were removed by immersing the substrate into the gold and chromium etchants. Finally, the chips were immersed into the photoresist remover and the silicone substrate was cut into two pieces, 5 × 5 mm each, using a laser micro miller.

The sensing layer was deposited by electrospinning the prepared solution on the silicone substrate. The setup of electrospinning included a high voltage power supply, a pipette/syringe tip, and a grounded collector plate. A syringe filled with the electrospinning solution was loaded into the syringe pump (KD Scientific), the positive electrode of the high voltage power supply (Gamma High Voltage Research) and the ground were attached to the adjustable substrate aligner (Appendix A). The electrospinning parameters were as follows: 8µL/min flow rate, 11kV DC, and a 5 cm distance between the nozzle and substrate. Appendix A shows the photo of the fabricated sensor.

After sensor fabrication, the effect of calcination and doping was studied. The processes involving calcination and doping are explained here: High-temperature calcination of the nanofiber layer increases the porosity of the sensing layer. A porous sensing layer facilitates the oxygen adsorption reaction at the surface of nanofibers by increasing the surface roughness and creating snowflake shape concavities. The calcination temperature plays a vital role in the enhancement of sensor sensitivity. The calcination temperature starts from the sensor operating temperature and goes up to 400 to 450 °C. The other factor that influences the sensor sensitivity is the calcination period. Cyclic heat treatment at intermediate temperatures is also another calcination method. In this paper, the calcination process was performed in the oven for 4 h at different temperatures (including 250, 300, 350, and 400 °C) to find the optimum calcination temperature. The Nicolet 6700 Fourier Transform Infrared (FTIR) spectrometer was used to examine the structure of the sensing layer. The FTIR analysis was carried out in the range of 4000 to 500 cm^−1^ with a scanning resolution of 2 cm^−1^. In addition, the morphology of the sensing layer was characterized using a scanning electron microscope (SEM).

Doping or adding metal nanoparticles to the metal oxide semiconductor sensing layer has been shown to enhance the sensitivity characteristics of a gas sensor. Among all the metals, a combination of gold with tin oxide has been shown to significantly increase the sensitivity of the sensor [28]. In this paper, gold doping of the sensing layer is performed by sputtering an ultrathin layer of gold on top of the semiconductor sensing layer. To ensure no contact between the gold electrodes and the ultrathin gold doping layer, an adequate amount of semiconducting material was placed before sputtering. The ultrathin gold layer was sputtered inside a clean room for three different thicknesses including 5, 10, and 15 nm to find the optimum gold doping layer.

The silicon wafer sensors were then integrated into a microfluidic channel to increase the selectivity of the sensor. The microfluidic channel was fabricated, similar to that of our previous work [28]. This microfluidic channel increases the selectivity of the sensor by taking advantage of varying diffusion rates of different gases. Slight alterations were required, as previous works [28] used a commercially bought sensor, whereas this study used an in-house fabricated sensor. The shape of the sensor compartment was altered to fit the square profile of the in-house sensor.

In order to compare the performance of the in-house fabricated sensor to commercial sensors, tests were performed with 1000 ppm methanol and THC/methanol (Sigma-Aldrich) using both the silicon wafer microfluidic sensor and commercial MOS-embedded microfluidic sensor (TGS 2602, Figaro Inc, Arlington Heights, IL, USA).

## 3. Results

### 3.1. Sensor Characterization

The characterization of the prepared sensing layer was carried out using scanning electron microscope (SEM) and FTIR spectrometry. Figure 2a,b shows the SEM image of the sensing layer after calcination for four hours at (a) 300 °C and (b) 400 °C, respectively. The FTIR spectrum for the film is shown in Figure 2c. The absorption peaks at 841 cm^−1^ (CH_2_ rocking) and 960 cm^−1^ (CC stretching and CH_2_ rocking), 1060, 1095, and 1145 cm^−1^ (COC stretching), 1241 and 1279 cm^−1^ (CH_2_ twisting), 1342 cm^−1^ (CH_2_ wagging), 1466 cm^−1^ (CH_2_ bending), 2888 cm^−1^ (CH stretching) were attributed to poly(ethylene oxide) (PEO) in the sensing layer.

After finding the optimum calcination temperature and gold doping thickness, the combined effect of these characterization methods is assessed by cyclic exposure of the sensor to 1000 ppm of methanol (Figure 3). The effect of calcination is more dominant as compared with gold functionalization (Figure 4). In order to ensure gold particles are incorporated into the composite, the sensing layer was characterized by the elemental analysis (Appendix A). The combination of doping and calcination provides the highest sensitivity towards methanol due to their synergetic effects. Porosity of the nanofibers combined with the presence of gold particles facilitates the adsorption of oxygen at the surface of the semiconductor sensing layer. Additionally, diffusion of the target gas molecules into the sensing layer (that has become a 3D structure after gold sputtering) makes the recovery step slower and causes a rise in the response during the last cycles of the exposure.

### 3.2. Sensor Response to THC

Appendix A shows the response of the sensor to different concentrations of THC in methanol. The calibration curve of the sensor was plotted and the sensitivity and limit of detection (LOD) of the sensor were calculated as 0.0008 V/ppm and 250 ppm, respectively (Appendix A). As the in-house fabricated sensor is placed at the end of the microfluidic channel and the concentration of the exposed THC solution at the surface of the sensor can be tuned by varying the exposure time of sensor to analyte, LOD of the sensor can be improved by increasing the exposure time of the sensor to THC.

Figure 5 compares the responses of the commercial sensor with those of in-house fabricated sensor for 1000 ppm of methanol. The commercial sensor shows better sensitivity towards methanol; whereas both responses have almost the same magnitude towards the THC/methanol mixture. Results demonstrate a difference in the response attributed to the analytes that the sensors are exposed to.

## 4. Conclusions

In this paper, a SnO_2_-based MOS sensor was fabricated using electrospinning. Pure and composite SnO_2_ nanofibers were obtained and calcinated at high temperatures as identified by SEM image. The performance of the sensor against methanol and THC was obtained. Then, the effects of the calcination temperature and the thickness of the Au doping on the performance of the sensor were studied. As observed from the results, the large surface area to volume ratio of the nanofibers followed by calcination at 350 °C for 4 h contributed significantly to the increase in the sensor’s sensitivity towards THC and methanol. Additionally, the effect of gold doping on electron absorptivity of the sensing layer, increases the sensitivity properties of the sensor. It was shown that 350 °C and 10 nm for calcination and Au-doping, respectively, are optimum parameters for detection of methanol. The porosity of the nanofibers combined with the presence of gold particles, facilitates the adsorption of oxygen at the surface of the semiconductor resulting in a raise in the response during exposure to the analyte. This paper introduces a novel technique to fabricate a microfluidic based gas sensor for THC detection.

## Figures and Tables

**Figure 1 micromachines-11-00190-f001:**
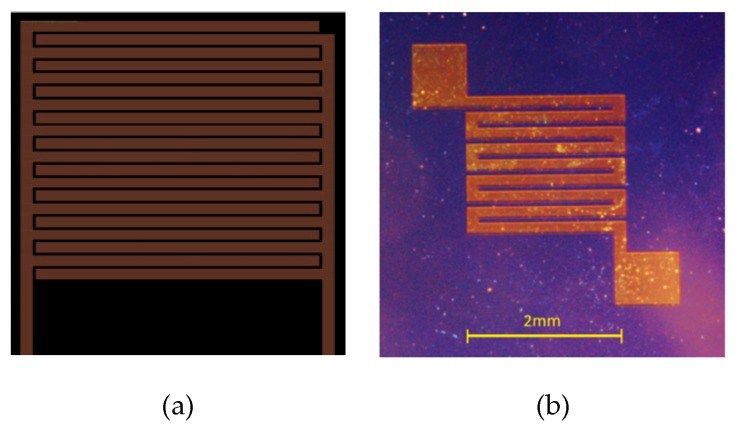
Representation of the microfabricated patterns on (**a**) top, interdigitated fingers; and (**b**) bottom, microheater.

**Figure 2 micromachines-11-00190-f002:**
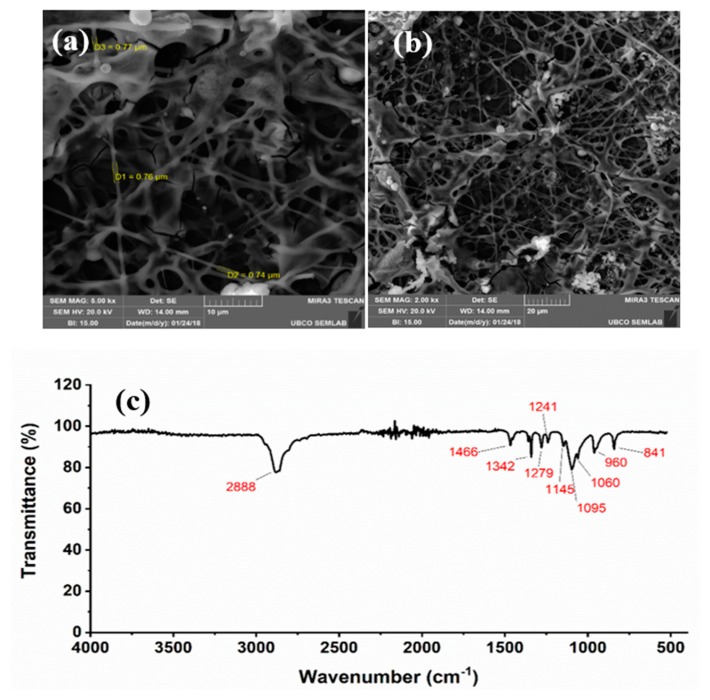
SEM images of the sensing layer after calcination for four hours at (**a**) 300 °C; (**b**) 400 °C; and (**c**) infrared spectra of the composite used as sensing layer.

**Figure 3 micromachines-11-00190-f003:**
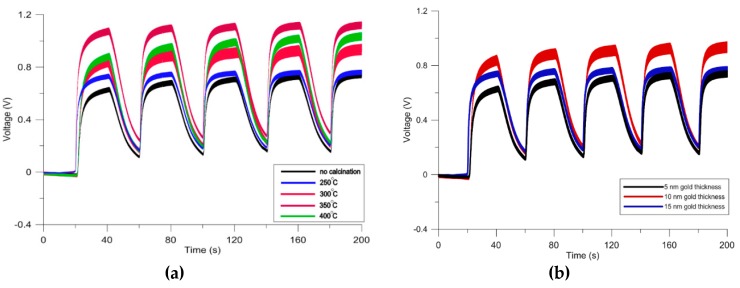
(**a**) Response of the sensors with different calcination temperatures to 1000 ppm methanol; (**b**) Test results of sensors with different thicknesses of deposited gold to 1000 ppm of methanol.

**Figure 4 micromachines-11-00190-f004:**
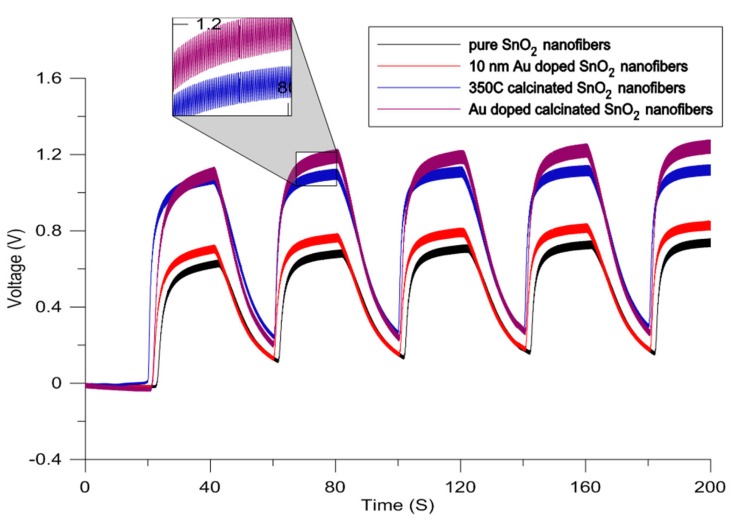
Effects of calcination, doping, and calcination-doping on the response of the sensor to 1000 ppm of methanol.

**Figure 5 micromachines-11-00190-f005:**
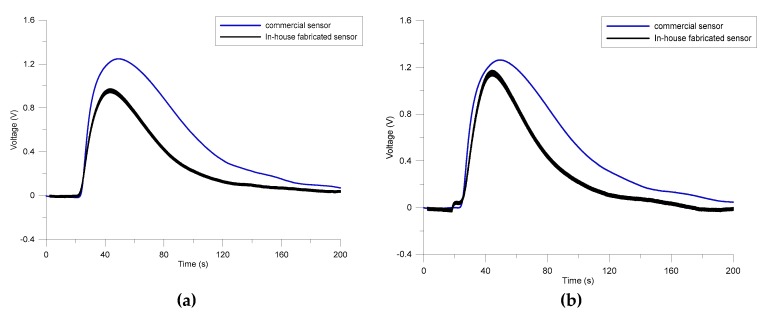
Comparison between the responses of the commercial and in-house built sensors towards 1000 ppm of (**a**) methanol and (**b**) methanol/THC mixture.

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
