# Peer review of "Fabrication of SnO2 Composite Nanofiber-Based Gas Sensor Using the Electrospinning Method for Tetrahydrocannabinol (THC) Detection"

_micromachines, 2020, doi:10.3390/mi11020190_

Round 1
Reviewer 1 Report
The authors report the fabrication of a nanocomposite of metal oxide (i.e. tin dioxide) and polyethylene oxide polymer for the development of the novel gas sensing system in this manuscript. The authors also improved the sensitivity of this sensing system via optimum calcination of nanocomposite and gold doping. Furthermore, the performance of this sensing system is better compared to that of a commercial sensor. The manuscript is well written. Therefore, I suggest the manuscript for publication in this journal after considering the following comments.
The authors should discuss more the novelty of work to the end of the introduction section. What is the selectivity, limit of detection and stability of their sensing system including in-house build sensor? A comparison table of previously published gas sensing systems and current works should be included. The elemental analysis (such as EDS and ICP) or high magnification TEM should be performed to confirm the incorporation of gold nanoparticles into nanocomposite. The authors should address the reason for the enhancement of sensitivity by calcination and doping in the revised manuscript.Author Response
Please see the attachment

Reviewer 2 Report
This research work presents a new sensor for THC detection. Overall, the topic is important, but lot of points must be improved before the paper can be further considered for publication such as :
1-The correlation between silicon patterned substrate and deposited nanofibers is totally not clear in the manuscript. It has to be clarified. A photo to the final sensor is recommended.
2- The sensor has to be tested at different ppm concentrations, not only at 1000. Then, the relation between concentration, voltage, time,... etc, should be analyzed. Also, the sensitivity relation between concentration and voltage has to be analyzed.
3-SEM images have to be mentioned in the paper core, not as supplemental files.
4-More characterizations may be added such as XRD, conductivity of thin film, and FTIR to support the properties of sensor surface morphology.
5-More literature about the importance of THC detection and comparison with current sensor are recommended.
Round 2
Reviewer 2 Report
All of the concerns of the reviewer have been clearly addressed by the authors. I would recommend to publish the paper in present form.